# Effects of *Lactobacillus rhamnosus* and *Enterococcus faecalis* Supplementation as Direct-Fed Microbials on Rumen Microbiota of Boer and Speckled Goat Breeds

**DOI:** 10.3390/vetsci8060103

**Published:** 2021-06-07

**Authors:** Takalani Whitney Maake, Olayinka Ayobami Aiyegoro, Matthew Adekunle Adeleke

**Affiliations:** 1Discipline of Genetics, School of Life Sciences, College of Agricultural, Engineering and Science, University of Kwazulu-Natal, Westville Campus, Private Bag X 54001, Durban 4000, South Africa; MaakeT@arc.agric.za (T.W.M.); AdelekeM@ukzn.ac.za (M.A.A.); 2Gastrointestinal Microbiology and Biotechnology, Agricultural Research Council-Animal Production, Private Bag X 02, Irene 0062, South Africa; 3Research Unit for Environmental Sciences and Management, North-West University, Potchefstroom Campus, Private Bag X 1290, Potchefstroom 2520, South Africa

**Keywords:** 16S rRNA, illumina sequencing, Boer goat, Speckled goat, probiotic, lactic acid bacteria, rumen

## Abstract

The effects on rumen microbial communities of direct-fed probiotics, *Lactobacillus rhamnosus* and *Enterococcus faecalis*, singly and in combination as feed supplements to both the Boer and Speckled goats were studied using the Illumina Miseq platform targeting the V3-V4 region of the 16S rRNA microbial genes from sampled rumen fluid. Thirty-six goats of both the Boer and Speckled were divided into five experimental groups: (T1) = diet + *Lactobacillus rhamnosus*; (T2) = diet + *Enterococcus faecalis*; (T3) = diet + *Lactobacillus rhamnosus* + *Enterococcus faecalis*; (T4, positive control) = diet + antibiotic and (T5, negative control) = diet without antibiotics and without probiotics. Our results revealed that Bacteroidetes, Firmicutes, TM7, Proteobacteria, and Euryarchaeota dominate the bacterial communities. In our observations, *Lactobacillus rhamnosus* and *Enterococcus faecalis* supplements reduced the archaeal population of Methanomassiliicocca in the T1, T2 and T3 groups, and caused an increase in the T4 group. Chlamydiae were present only in the T5 group, suggesting that probiotic and antibiotic inhibit the growth of pathogens in the rumen. We inferred, based on our results, that *Lactobacillus rhamnosus* and *Enterococcus faecalis* favour the survival of beneficial microbial communities in the goats’ rumen. This may lead to an overall improved feed efficacy and growth rate.

## 1. Introduction

Goats are raised for milk, meat, cheese, skin and hair, and play an important part in religious and cultural ceremonies in South Africa. The country has a successful goat industry consisting of commercial and indigenous goat breeds [1]. The Boer, Savannah and Kalahari are commercially developed breeds that have turned out to be important worldwide [2]. These South African goats are recognized for their rapid growth and good meat carcass traits [3]. Goat as a ruminant have the ability to break down plant material through fermentation in the rumen using anaerobic microbiota such as bacteria, fungi and protozoa, which convert feeds into energy [4]. In goats, and in other ruminants, the rumen microbial diversity and the host physiology can be manipulated by diet, even though the microbial community is mostly stable throughout the animals’ life [5]. Successfully, antibiotics are in use to enhance beneficial gut microbial diversity. However, the long-lasting use of antibiotics has led to worldwide concerns of antibiotic-resistant microorganisms, which pose threats to human health and the environment [6]. As a result, the use of antibiotic as feed additives has been banned in most European Union Countries since 2006. Alternatives, including probiotics, are possible replacement for antibiotics [6]. Previous studies have shown that the use of probiotics, such as lactic acid bacteria, can improve nutrient digestibility, decrease pathogen colonization in the gut, improve balance in the gastrointestinal microbiota and enhance ruminants’ health and productivity [7].

In recent years, metagenomics analysis has provided more information about taxonomic diversity and interactions of the gut microbiomes. More information about microbiomes can provide insights on rumen microbial communities and possible applications in animal husbandry. Studies have shown that ruminants can adapt to new diets and provide and provided understanding of how the intestinal microbiome interact and contribute to the well-being of the animal [8]. As microbiome communities are of great importance in the breakdown and absorption of nutrients, it is important to determine the effect of direct-fed microbes on the rumen microbiota [9]. Therefore, in this study, we explored the microbial diversity and composition in the rumen of Boer and Speckled goats, under the same feeding regimen, supplemented with *Lactobacillus rhamnosus* and *Enterococcus faecalis*, as putative probiotics.

## 2. Materials and Methods

### 2.1. Animals, Treatments and Sampling

All animal experimental procedures were performed under protocols approved by the Agricultural Research Council-Animal Production Institute Ethics Committee (APIEC17/23), before the commencement of the trial. The trials were done at the GI Microbiology and Biotechnology unit, and the Small Stocks Unit in Irene, of the Agricultural Research Council Animal Production Institute, Gauteng Province. The Agricultural Research Council, Irene campus is located at 25°55′ South and 28°12′ East.

The lactic acid bacteria (putative probiotics) used in this study were isolated and characterized from fresh faecal samples of indigenous veld goats (IVG). These goats are known to adapt in harsh environmental conditions and have resistance against parasites and diseases [1]. Molecular sequencing and probiotic and technological properties such as antimicrobial activity, acid, and bile tolerance were used to characterise and identify the two potential probiotics. The potential probiotic bacteria were prepared on De Man Rogosa and Sharpe (MRS) broth (Oxoid, Basingstoke, Hampshire, England) anaerobically, and preserved in 25% glycerol in the ultra-low freezer. The two putative probiotics were revived by inoculation in MRS broth. For suspension, MRS broth was inoculated with, 1% (*v*/*v*) culture and incubated anaerobically at 37 °C overnight prior to administering.

Goats were treated in accordance with the established standards for the use of animals’ ethical guidelines. The goats were vaccinated (CDT Vaccine) against *Clostridium perfringens* type C and D (overeating disease) and *Clostridium tetani* (tetanus) 15 days before the start of the trial to control diarrhoea. A total of thirty-six randomly selected goats, average age 25 weeks old, including Boer and Speckled, were used for this trial. The trial lasted for 30 days after an initial 30 days of adaptation. The body weights of goats at the beginning of the trial were: Boer males (15.8 ± 2.6 kg), Boer females (14.7 ± 1.2 kg), Speckled males (14.4 ± 3.9 kg), and Speckled females (14.2 ± 5.1 kg). The goats were separated per treatment according to breed (treatment 1 = 4 Boer and 4 Speckled, treatment 2 = 3 Boer and 3 Speckled, treatment 3 = 3 Boer and 3 Speckled, treatment 4 = 4 Boer and 4 Speckled, treatment 5 = 4 Boer and 4 Speckled) and sex (treatment 1 = 4 males and 4 females, treatment 2 = 2 males and 4 females, treatment 3 = 3 males and 3 females, treatment 4 = 4 males and 4 female, treatment 5 = 4 males and 4 females) into the trial shelters. The five experimental treatments were as follows: (T1) = diet + *Lactobacillus rhamnosus*; (T2) = diet + *Enterococcus faecalis*; (T3) = diet + *Lactobacillus rhamnosus* + *Enterococcus faecalis*; (T4, positive control) = diet + antibiotic; and (T5, negative control) = diet without antibiotics and without probiotics. The diet used was in the form of pellets to provide nutrient requirements regardless of the treatment, as recorded in Table 1. Antibiotic lincospectin was added to the diet in the positive control (T4) group. Freshwater and hay were provided ad libitum for all the goats. The weekly administration of probiotics to goats was done orally using a dosing gun at a dosage of 5 mL of 2 × 10^9^ cfu/ML of fresh live culture per head, repeated every week at 08:00 am for four weeks. The goats were weighed individually before and after the trial using a calibrated weighing scale.

Ruminal samples were collected from all the goats at the beginning of the trial and on the last day of the trial using the ororuminal collection method [9]. About 100 mL of ruminal fluid samples were collected before and after the trial at 07:30 am before feeding on day 1 and day 30 of the trial, by inserting a sterilized tube to the stomach through the mouth of the goat. A volume of 40 mL of the collected rumen content was kept on ice and transferred to 50 mL centrifuge tubes and centrifuged at a speed of 10,000× *g* for 15 min. The collected supernatants were transferred into other clean and sterile tubes and stored immediately at −80 °C until DNA extraction. The pH of the ruminal fluid was measured using a pH meter immediately after collection. Results are shown in Table 2, adopted from our previously published article [10].

### 2.2. DNA Extraction, PCR Amplification and MiSeq Sequencing

Total genomic DNA was extracted from rumen fluid samples using PureLink Microbiome DNA Purification Kit (Thermo Fisher, Johannesburg, South Africa) according to the manufacturer’s instructions. The quantity of the DNA was assessed using Qubit 4 Fluorimeter (Invitrogen, Johannesburg, South Africa). The extracted DNA samples were used as templates for amplifying the V3-V4 region using the following primers, which include Illumina overhang adapter sequences [11]:

16S Amplicon PCR Forward Primer = 5′TCGTCGGCAGCGTCAGATGTGTATAAGAGACAGCCTACGGGNGGCWGCAG3′, 16SAplicon PCR Reverse Primer = 5′GTCTCGTGGGCTCGGAGATGTGTATAAGAGACAGGACTACHVGGGTATCTAATCC3′. The PCR reaction was carried out as follows: 2.5 μL microbial genomic DNA (5 ng/μL), 5 μL of amplicon reverse primer (1 μM), 5 μL of amplicon forward primer (1 μM), 12.5 μL of 2XKAPA HiFiHotStart Ready Mix (KAPA Biosystems, Cape Town, South Africa), with the following conditions on the thermal cycler: initial denaturation at 95 °C for 3 min, 25 cycles (95 °C for 30 s, 55 °C for 30 s 72 °C for 30 s) and a final extension at 72 °C for 5 min. Amplicons were visualized using agarose gel electrophoresis. A band of size 550 bp was excised from the gel and purified using NucleoSpin Gel and PCR Clean-up kit (Macherey-Nagel, Valencienner, Düren, Germany) according to the manufacturer’s instructions.

Illumina MiSeq library preparation and sequencing was carried out at the ARC-Biotechnology Platform, South Africa, and the raw data generated submitted to the National Center of Biotechnology Information (NCBI) Sequence Read Archive (SRA) database under Bio Project: PRJNA579264.

### 2.3. Data Analysis

Trimmomatics version 0.36 was used to trim raw data sequences that had been generated from Illumina sequencer MiSeq in order to remove adapter sequences. PANDAseq was used to merge the trimmed reverse and forward reads. The merged sequences were imported to Qiime2 for analysis. DADA2 was used to remove chimeras from the imported sequences.

The Green Gene database was used to perform OTU picking. The R studio was used to carry out further analysis on taxonomic classification and diversity. Alpha diversity of samples was calculated using three indices: Shannon index, Simpson index and Chao1 index. For multivariate analysis, non-metric Multidimensional Scaling Plots (NMDS) were calculated based on Bray-Curtis Dissimilarity distances.

## 3. Results

### 3.1. OTU Clustering and Taxonomic Annotation of the Goat Rumen Microbiome

To better understand the OTU information and their taxonomic annotation, tags and OTU were calculated. The taxonomic classification between Boer and Speckled was compared, showing dominant phyla and genera for both the sampling period (Figure 1A). Microbial abundance at the species level was also evaluated by comparing the abundance between day 1 and day 30 of the trial (Figure 1B).

Using the OTUs, a Venn diagram (Figure 2A) was created to show the number of OTUs shared between the goat breeds. The number of mutual OTU in rumen samples between two breeds was 3251, representing 59% of shared OTU, while Speckled goats had 23% and Boer goats 17% (Figure 2A). The most frequently abundant 36 OTUs were observed among all the treatments (Figure 2B). The distribution patterns showed that core OTUs may perform same basic functions among the five treatment groups.

### 3.2. Bacterial and Archaeal Composition

The taxonomic classification resulted in naming of 19 phyla, 28 classes, 39 orders, 72 families, and 97 genera across bacteria and archaea domains.

Chloroflexi in Treatment 1 group, Fusobacteria in Treatment 2 group, SR1 in Treatment 3 group, WS6 and Verrucomicrobia in Treatment 4 group, and Fusobacteria in Treatment 5 group were observed to be more enriched on day 30 of the trial in the rumen microbiota of the experimental animals (Figure 3A).

At the genus level, the most predominant genera were Prevotella, Anaerofustis, Clostridium, Fibrobacter, and Martelella (Figure 3B). The most predominant bacterial species included *Prevotella ruminicola, Clostridium aminophilum, Fibrobacter succinogenes*, and *Clostridium clostridioform* (Figure 3C).

Across all treatment groups, the archaeal community was dominated by Methanobrevibacter, followed by vadinCA11, Methanoplanus and Methanosphaera. The genus vadinCA11 decreased in all the treatment group except the treatment group 4 (Figure 3D).

### 3.3. Comparison of Bacterial Diversity

Alpha diversity was used as a measure of diversity within rumen microbiota. Alpha diversity of gut microbiota was shown to be influenced by breed, sex, gender and treatment. Three indices were determined (Shannon, Simpson, and Chao1) (Figure 4). All three indices showed an increment on the final day in treatment groups (1 to 4). Treatment 5 only showed an increment in Chao1 index. The differences were consistent in Shannon index and Chao1 indexes across the five treatment groups at the subsample depth point. Rumen microorganism present in Treatment 4 group (H = 2.4) had a higher Shannon index than that of Treatment group 1 (H = 2.0), 2 (H = 2.05), 3 (H = 1.90) and 5 (H = 2.21). The values of the three indexes were significantly higher in Treatment group 4 as compared to other treatments, indicating that the alpha diversity of rumen microbiome was higher in treatment group 4. Significant difference in alpha diversity was observed between Boer and Speckled goats. Speckled goats had higher Shannon and Simpson indexes (Figure 5).

Samples were found to be dispersed according to treatment groups. The same pattern was observed across all NMDS among breeds and genders as presented in Figure 6.

Analysis of similarity (ANOSIM) showed that there were many similarities in microbial composition in the rumen across the treatment 1, 2, 4 and 5 (Figure 7A). ANOSIM also showed that no differences were observed in the rumen microbial structure between Boer and Speckled goats (Figure 7B).

## 4. Discussion

In recent year, the association between the rumen microbial community and the host has revealed to have a significant effect on the host’s well-being [12,13]. Several studies have shown that lactic acid bacteria have favourable effects on the host [14,15,16]. In the present study, we evaluated microbial diversity and composition of five treatment groups of Boer and Speckled goat breeds. The rumen microbiota was altered by the supplementation of antibiotics and probiotics. A decrease in the ruminal pH was also observed in all the treatment groups. Ruminal pH plays a significant role in maintaining the internal balance in the rumen environment; therefore, it is important to maintain a moderate stable pH for ruminal fermentation. Franzolin and Dehority [17] also observed a decrease in pH related to diet and its importance to the stability of the gut microbiota.

In both breeds, a microbial diversity of 19 bacteria phyla (5470 OTUs), was observed. A rarefaction curve was constructed to show that the sequencing depth of the sample was sufficient (Appendix A). When compared to other studies, this study illustrated the high abundance of bacteria in the rumen of goat’s breeds at various taxonomic levels [5].

Irrespective of dietary treatment groups, Bacteroidetes, Firmicutes and Proteobacteria were found to be the dominant phyla across all the treatments, with a high abundance of Ruminococcaceae, Prevotellaceae, Lachnospiraceae and low abundance of Veillonellaceae and Bacteriodaceae. These result were in agreement with previous work by Wang et al. [18], who identified microbial using Next-generation sequencing technique from goats. Furthermore, Cremonesi et al. [19], also found similar results. In this study, Prevotellaceae was identified as the dominant bacterial family in all the treatment groups in day 1 and day 30. Bacteroidaceae and Prevotellaceae are known to be the main families that plays a vital role in the degradation of the feed in goats [13]. Archaea, accounts for about 4% of the ruminal microbes [20] and Methanobrevibacter was the main genus, confirming what previously reported studies on rumen from goat [19,21], sheep [21], and cattle [22]. Methanobrevibacter is a genus from Methobacteriaceae, which are obligate anaerobes that produces methane as a major catabolic product [23,24].

The observed increment of some of bacteria in the rumen may result from increased microbes, which breakdown carbohydrates and fibre ingested by the animal. The presence of species (*Ruminococcus callidus, Fibrobacter succinogenes* and *Clostridium* spp.) which promote degradation of cellulose into soluble carbohydrates [24], were observed in high abundance in all treatment groups (Figure 3C). Correspondingly, Bacteroidales were more abundant in all treatment groups; however, the number decreased during the 30 days of the experiments (Figure 1B). An increase of Lactobacillaes at day 30 was observed compared to day 1 of the trial (Figure 1B). Shabana et al. [25], also recorded an increase in Lactobacillus as the age of the goat increased.

The increased abundance of the genera Lachnospiraceae and Bacillus in the negative control group (Treatment 5) in the gut may be due to an undisturbed ecosystem. The genus Bacillus is a genus characterized by high proteolytic activity [26]. Whereas, members of the family Lachnospiraceae are associated with butyrate production through carbohydrate digestion [27,28]. Although no increase in Lachnospiraceae was observed in the treatment groups, a high abundance was observed in all treatment groups.

The genus vadinCA11 from the order Methanomassiliicocca, which was initially high in all the treatment groups at day 1 of the experiment, decreased (not significant) after 30 days of experimental trials in all the treatments groups except in Treatment 4 group (the antibiotic group) (Figure 3D). The presence of this genus has the potential to allow the microbiome to adapt quickly to environmental stress like diet changes. However, the abundance of this genus must be controlled because it can produce additional ammonium through methanogenesis [28]. Therefore it is of great importance to include feed which decreases methane production [22] without affecting fermentation and fibre degradation [28].

The presence of fibrolytic bacteria, *Fibrobacter succinogenes*, a fibre-degrading bacteria was also observed (Figure 3C). Fibrolytic bacteria are important in the ruminal production of propionate [29].

Alpha diversities within treatments revealed that microbial diversity was altered with an increase in the richness and overall diversity of the bacterial species observed with Treatment 4 (positive control), followed by Treatment 3 (combination of *Lactobacillus rhamnosus* and *Enterococcus faecalis*). Breed variation also affected diversity, as Speckled goats had higher Shannon index and Simpson index values than Boer goats. These findings show that the host genotype plays a significant role in maintaining the rumen microbial structure and functions. The results are in accordance with other studies that investigated rumen microbial diversity in cows [29,30], sheep [25,31] and goat [32].

Beta diversity showed no significant dissimilarities between the Boer and Speckled goat breeds, and also between treatment 1, 2, 4 and 5, which could mean that there was no distinct diversity in the rumen microbiota of the treatment group. This result was also supported by an Adonis plot (Figure 7). The R value was 0.055. The closer the R-value is to 1, the greater the difference between the treatments. Noel et al. [33] also observed that diet had no significant effect on dissimilarities between microbial communities.

## 5. Conclusions

Our study showed that the administration of lactic acid bacteria as putative probiotics slightly altered rumen microbial structure and abundance. Although there were some variations in microbial communities between treatments, similar rumen phyla (Bacteriodetes, Actinobacteria, Firmicutes, Tenericutes and Fibrobacter) were abundant in all the treatment groups. The observed rich and diverse microbiome could be the effect of direct-fed microbials to maintain the balance of gut microbiota, and hence, the well-being of the animal. In addition, breed variation had an effect on microbial composition and structure of the rumen environment in goats from our study. We inferred, based on our results, that *Lactobacillus rhamnosus* and *Enterococcus faecalis* favour the survival of beneficial microbial communities in the goats’ rumen, and this may lead to an overall improved feed efficacy and growth rate.

## Figures and Tables

**Figure 1 vetsci-08-00103-f001:**
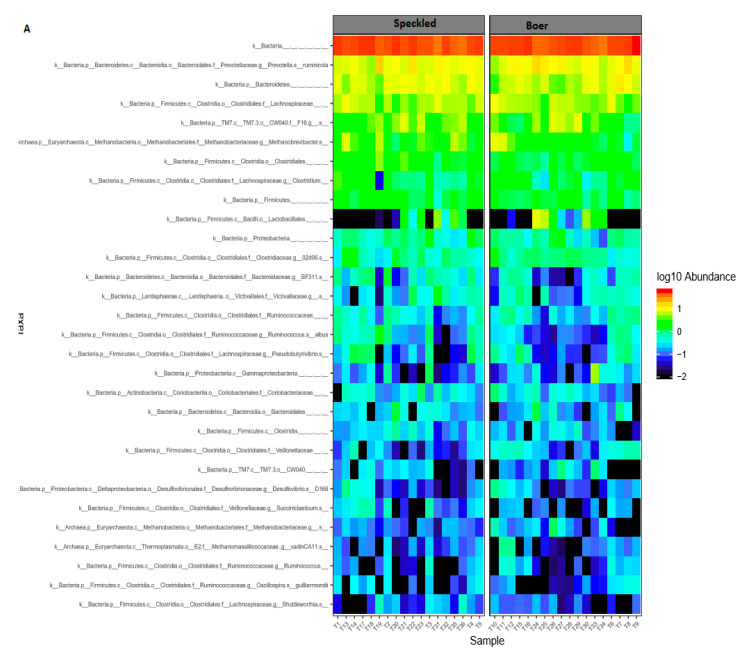
Relative abundance of bacterial and archaeal taxonomic classification between two goat breeds (**A**). Comparison of top 30 most abundant taxa between day 1 and day 30 of the trial (**B**). T1-T18 indicate samples collected at the beginning of the trial and T19-T36 was collected at the end of the trial. Relative abundance values for each taxonomic classification is illustrated by color intensity according to the legend provided on the scale of −2 to 1.

**Figure 2 vetsci-08-00103-f002:**
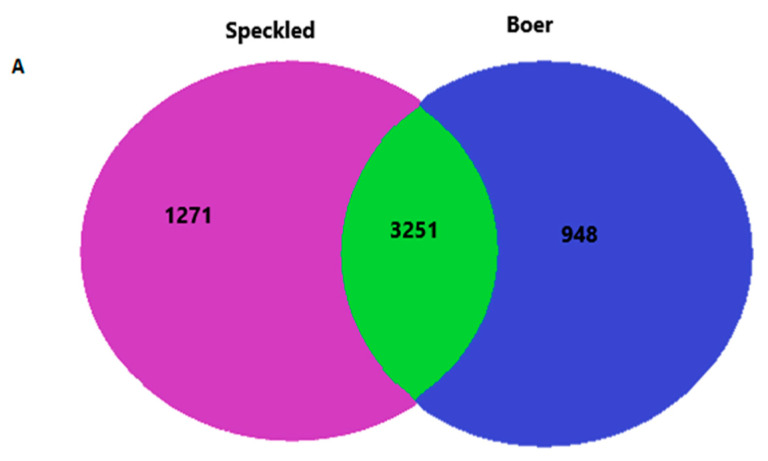
Venn diagram of number of operational taxonomic units of bacteria at day 30 of the trial between two goat breeds (**A**) Speckled (purple) and Boer (blue), and five treatment groups (**B**): treatment 1 (blue), treatment 2 (yellow), treatment 3 (orange), treatment 4 (green), treatment 5 (purple). The numbers in the diagrams represent how many OTUs were unique in the five treatment groups or shared (similar) between sections as their areas overlaps.

**Figure 3 vetsci-08-00103-f003:**
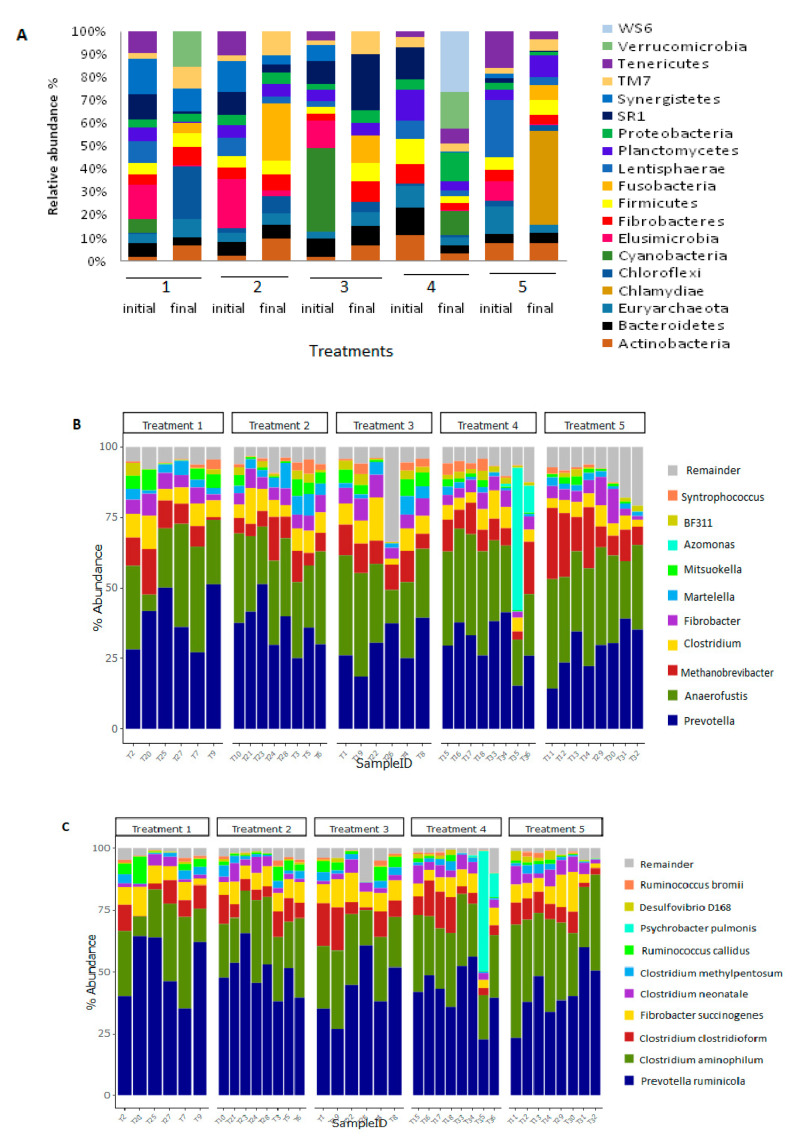
Relative abundance of microbial communities across the five treatment groups of the trial at (**A**) phylum, (**B**) genus, (**C**) species levels. The abundance at genus level was also shown (**D**). “Remainder” includes all phyla or genera with less than 1% relative abundance. T1–T18 indicate samples collected at the beginning of the trial and T19–T36 were collected at the end of the trial. Each bar represents the average relative abundance of each bacterial or archaeal taxon within a group.

**Figure 4 vetsci-08-00103-f004:**
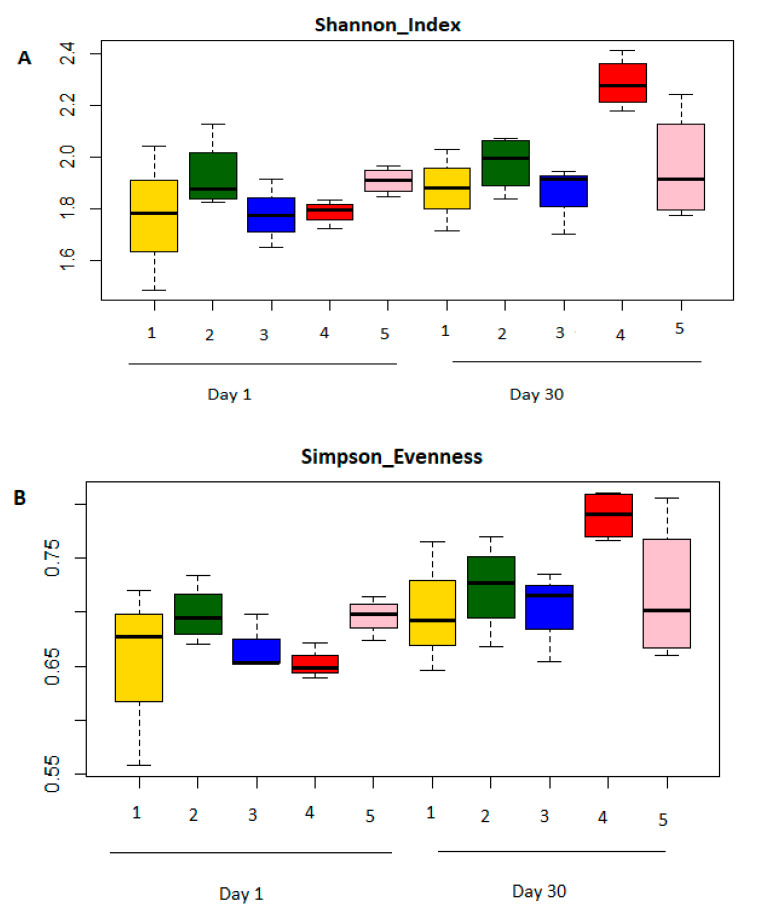
Alpha diversities within each treatment group at day 1 and day 30. Yellow (treatment 1), green (treatment 2), blue (treatment 3), red (treatment 4) and pink (treatment 5). Three indices were measured: Shannon index (**A**), Simpson evenness (**B**) and Chao1 (**C**). The top and bottom boundaries indicate the 75th and 25th quartile values, respectively. The horizontal lines within each box represent median values.

**Figure 5 vetsci-08-00103-f005:**
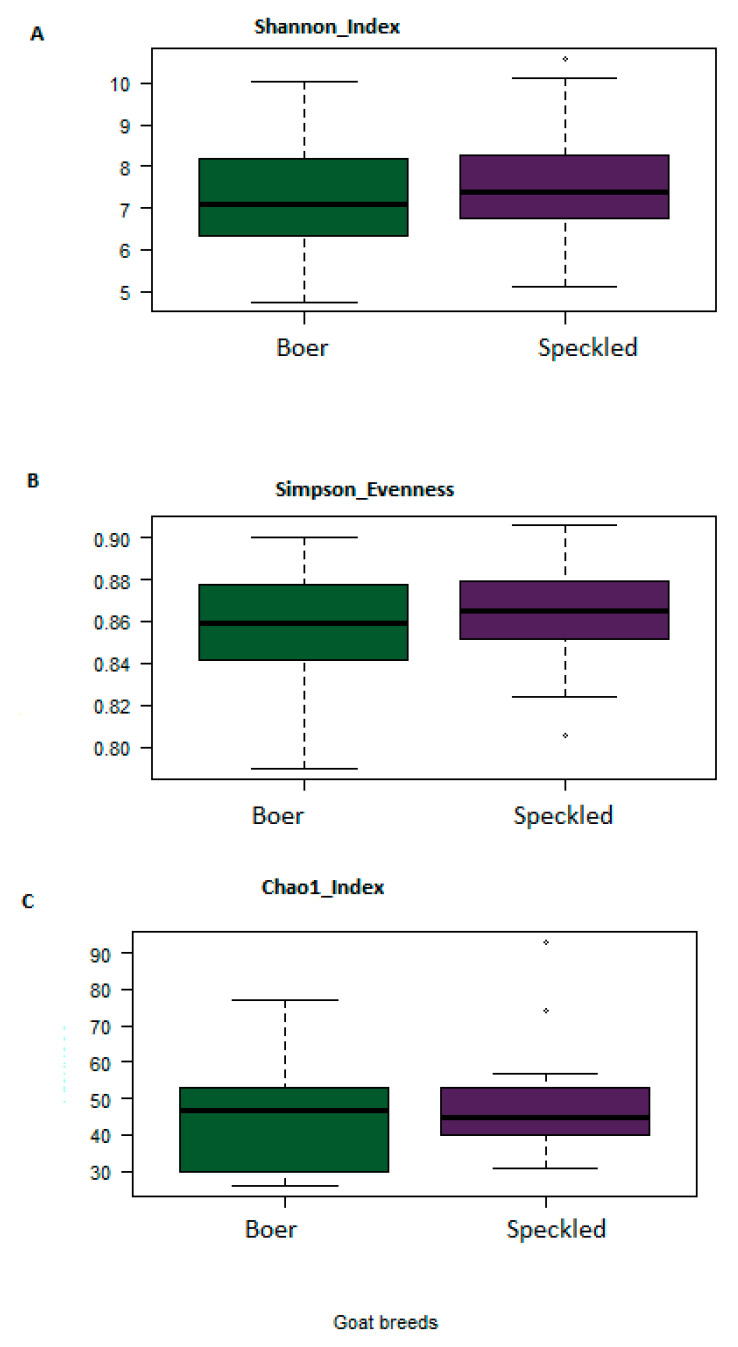
Alpha diversities within Boer and Speckled goats on day 30 of the trial. Blue (Boer), and purple (Speckled). Three indices were measured: Shannon index (**A**) Simpson evenness (**B**) and Chao1 (**C**). The top and bottom boundaries indicate the 75th and 25th quartile values, respectively. The horizontal lines within each box represent median values.

**Figure 6 vetsci-08-00103-f006:**
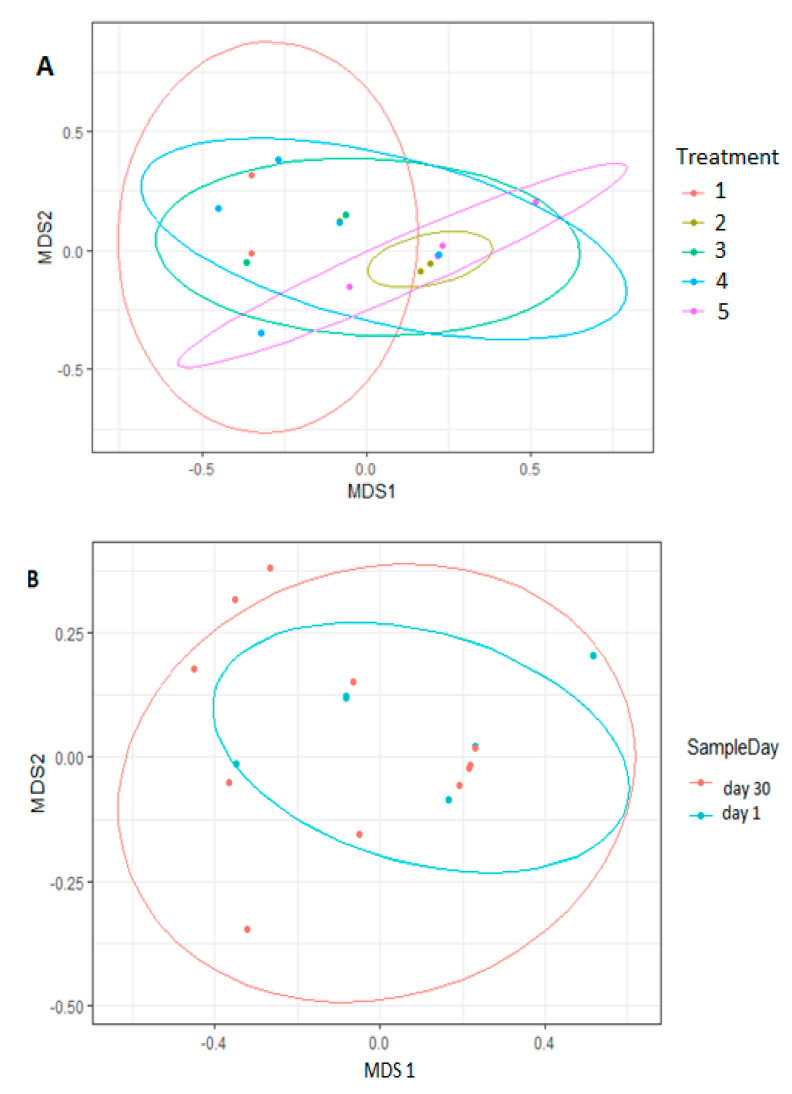
Nonmetric Multidimensional Scaling Plots (NMDS) based on Bray-Curtis Dissimilarity distances in rumen content of goats treated with *Lactobacillus rhamnosus* (Treatment 1), *Enterococcus faecalis* (Treatment 2), combination of *Lactobacillus rhamnosus* and *Enterococcus faecalis* (Treatment 3), antibiotic (Treatment 4), and negative control (Treatment 5). Each point represents sample and the colours represent; treatment (**A**), sample day (**B**), breed (**C**), sex (**D**).

**Figure 7 vetsci-08-00103-f007:**
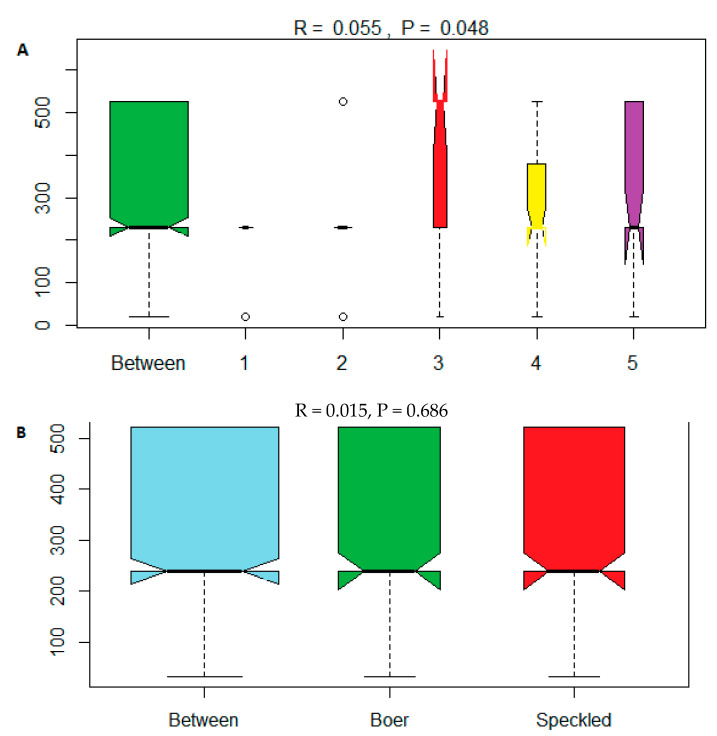
Adonis plots showing similarities between treatments. Analysis of similarities (ANOSIM) of the differences in structure of bacterial community in rumen of goats between treatments (**A**) 1 = (Treatment 1); 2 = (Treatment 2); 3 = (Treatment 3, red); 4 = (Treatment 4, orange); 5 = (Treatment 5, purple). The difference in structure of bacterial community between Boer and Speckled goats were also shown (**B**). Y axis shows the ranks of dissimilarity. The ends of the whiskers represent the minimum and maximum of all the data within the group. “Between” represents the difference between the five treatment groups, the closer the R-value is to 1, the greater the difference between the breeds and treatment.

**Table 1 vetsci-08-00103-t001:** Nutrient composition of the commercial diet.

Nutrients	g (kg)
Protein	150
Fat	25
Fibre	110
Calcium	8
Phosphorus	2
Urea	1
Chloride	9
Sodium	9
Magnesium	1
Potassium	6

**Table 2 vetsci-08-00103-t002:** Effect of breed and treatment on ruminal pH of goat (Adopted from Maake et al., 2021 [10]).

Parameter	T1	T2	T3	T4	T5	Boer	Speckled	*p*-Value
Initial pH	6.99 ± 0.44	6.56 ± 0.42	7.12 ± 0.41	7.5 ± 0.45	7.19 ± 0.43	7.12 ± 0.42	7.12 ± 0.42	0.57
Final pH	6.32 ± 0.41	6.37 ± 0.46	6.18 ± 0.52	6.4 ± 0.52	6.36 ± 0.56	6.80 ± 0.53	6.34 ± 0.55	0.0001

## Data Availability

Raw reads generated by MiSeq Illumina sequencer were deposited to the National Center of Biotechnology Information (NCBI) Sequence Read Archive (SRA) database under BioProject: PRJNA579264. Data on request from the corresponding author.

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
