# Peer review of "Effects of Lactobacillus rhamnosus and Enterococcus faecalis Supplementation as Direct-Fed Microbials on Rumen Microbiota of Boer and Speckled Goat Breeds"

_vetsci, 2021, doi:10.3390/vetsci8060103_

Round 1

Reviewer 1 Report

I have checked the manuscript carefully and i found that 51% similarity from other sources according to international requirement the similarity is too high and in this condition i can not suggest this manuscript for publication, i have attached the similarity report kindly find attachment. 

Author Response

Comments from reviewer 1:

Comments and Suggestions for Authors: I have checked the manuscript carefully and i found that 51% similarity from other sources according to international requirement the similarity is too high and in this condition, I cannot suggest this manuscript for publication, i have attached the similarity report kindly find attachment.

The manuscript is now reworked, and submitted for Turnitin Plagiarism Check, this gave 23% similarity index. See the attached report, most of the similarities are from our previous published work and submitted thesis in terms of affiliations, methodologies, names, etc.

Reviewer 2 Report

General comments:

The manuscript has been improved, however some changes are necessary.

Abstract:

Line 19, 22 Use italics to scientific names such as “Enterococcus”. Here and in all the manuscript.

Introduction:

Line 33 Delete “dairy”. “Other products” include skin, hair, cheese, etc.

Line 38 I think that the right text is: “…to breakdown indigestible plant material in monogastrics through fermentation…”, because indigestible in ruminants, is practically indigestible.

Material and methods:

Line 86 The word “against” is repeated. Please, delete one.

Line 108 “each occasion”: What times At what time of the day? How many hours before or after feeding? How many collections?

Line 109 “40 ml were collected to DNA extraction” and What happened to the other 60ml?

Discussion:

Line 311, 313, 333, 367 Change “and co-workers” instead “et al.”.

Line 322 Change “feed” instead “food”. The word "food" is used when talking about human food.

Lines 352-353 Is this sentence correct? Fibrolytic bacteria are not more correlated with acetate production?

Author Response

See the attached document

Reviewer 3 Report

Dear author. I do see an interesting research whoever some important points in manuscript needs to be reviewed e change.  The tittle should be more specific. The author analyzed the rumen microbiota. It should be more specific. How the author, can explain the reason for to use the antibiotic as a positive control? It is hard to compare with the other treatments. The way of the analysis is showed on the figure 2 and 3 is hard to see the shift of the abundance between the microbes. We do not know which sample they are, if they are from before or after the treatments. There is no p-value The conclusion is not in accordance with the results showed

L14 - Microbial community from where? It should be more specif. Even in whole manuscript.

L37- Microbiota from where? The gastrointestinal tract compartments from ruminants have a different microbiota with different function as well.

L64-66- Please specify better the objective of work, it’s not clear. Where is the effect?

L89- Please correct the breed goat

L95- Include the breed and gender number per treatment

L127-128- Include the primer reference.

L127-128 – How do you know that this primer works for archaea as well?

L167- Results 3.1- It is not clear from which period belong the samples from figure 1. It is before or after the treatment? What means “sampling days” in the legend. It is really confused

L184- The same as before the figure 2a is from before or after the treatment? Specified better the legend.

L221-222- Is not shown the Verrucomicrobia and Tenericutes on the figure 3a.  Is statically significant?

L231- The author already said this on the line 229

L235-236- It is not correct. Based on the figure 4a It seems that the group 5 did not had an increased for Shannon and Simpson index.

L235-242- The difference founded in alpha diversity are significant? This was not indicated on the text nether on the figure F4 A-C

L167-185- How can you compare the rumen microbiota between two breeds if they received a different treatment?

 L243- Dietary? I think it should be treatment.

L271- Replace rumen contents for rumen microbiota

L274- “Gut microbiota of a goat”? It is seeming like a sample. Please replace this phrase

L289- Where is the p-value?

L294: Microbial abundance and diversity from where? Please add structure as well

L296- Ph value was decreased in which treatment? It is not specified

L301-302- Diversity? This sentence is wrong and really confused

L304- Where is the rarefaction curve?

L308- Add goats rumen microbiota

L315-316: How the author can say that? There is no digestibility assay on the work

L321- Replace gut for rumen

L322- Composition or structure?

L321-324- This sentence is confused. The samples did not clustered separated, as showed on Fig 4.

L331- There is no figure 2c

L333- I do not see lactobacillaes on figure 1B

L352-There is no figure 2C. I do not see an increase of fibrobacter on none of the treatments.

Round 2

Reviewer 1 Report

Please improve the English quality throughout the manuscript kindly write the manuscript in scientific way not in a common way thanks 

Author Response

The English Language is now extensively revised in the manuscript as suggested.

Thank you.

Reviewer 2 Report

General comments:

 The manuscript has been improved. The authors did a good job and I am satisfied with the results.

Author Response

The reviewer acknowledged the through revision of the manuscript. So, no comments to respond to.

Reviewer 3 Report

line 194- On figure 3, or you add a title in all graphics or you take off

Line 202-203- I do not see increasing of Cyanobactera and Bacteroidetes after 30 days of trial on figure 3, in any of the treatments

Line 204- "abundance of Verrucomicrobia and Tenericutes were more enriched in the rumen microbiota of Treatment 4 group" . Is this significant?

line 208- Rumminococcus callidus and Fibrobacter succinogenes are not the most predominat 

line 225- This is not beta diversity

Line 228- The species name should be italic

Line 232- Correct (Figure 4)

line232- Add the significance indication on figure 4

line 315- Ruminococcus succinogenes ?

line 326-327- The sentence is confuse. 

line 239- Add significance indication on figure 5

line 319- "Correspondingly, Bacteroidales was found in more abundance in all treatment groups; however, the levels decreased through days". FIGURE 1?

line 340- The phrase  "showing higher diversity than other treatment groups" was already said before

line 341- "followed by Treatment 3 (combination of Lactobacillus rhamnosus and Enterococcus faecalis)" Is this significative? If is not you need to especify

Author Response

Responses to Reviewer 3 Comments:

We appreciate the reviews of our article and conditional acceptance for publication in your journal after minor revisions.

We have attempted to respond to all of the reviewer’s comments and suggestions, unless where the comment is not clear to us and we have indicated those in our responses.

We hope that you found these in order.

Thank you.

line 194- On figure 3, or you add a title in all graphics or you take off

Titles are now removed.

Line 202-203- I do not see increasing of Cyanobactera and Bacteroidetes after 30 days of trial on figure 3, in any of the treatments

The statement is now rewritten as thus: Chloroflexi in Treatment 1 group, Fusobacteria in Treatment 2 group, SR1 in Treatment 3 group, WS6 and Verrucomicrobia in Treatment 4 group, and Fusobacteria in Treatment 5 group were observed to be more enriched on day 30 of the trial in the rumen microbiota of the experimental animals (Fig. 3a).

Line 204- "abundance of Verrucomicrobia and Tenericutes were more enriched in the rumen microbiota of Treatment 4 group" . Is this significant?

The statement is now rewritten as thus: Chloroflexi in Treatment 1 group, Fusobacteria in Treatment 2 group, SR1 in Treatment 3 group, WS6 and Verrucomicrobia in Treatment 4 group, and Fusobacteria in Treatment 5 group were observed to be more enriched on day 30 of the trial in the rumen microbiota of the experimental animals (Fig. 3a).

line 208- Rumminococcus callidus and Fibrobacter succinogenes are not the most predominat 

Replaced with Prevotella ruminicola, Clostridium aminophilum, Fibrobacter succinogenes, and Clostridium clostridioform (Fig. 3c).

line 225- This is not beta diversity

Now changed to alpha diversity

Line 228- The species name should be italic

Now written in italics

Line 232- Correct (Figure 4)

Now changed to Figure 4

line232- Add the significance indication on figure 4

This comment is not clear to us

line 315- Ruminococcus succinogenes ?

Replaced now with Fibrobacter succinogenes

line 326-327- The sentence is confuse. 

The sentence now rephrased as : The genus vadinCA11 from the order Methanomassiliicocca, which was initially high in all the treatment groups at day 1 of the experiment, decreased after 30 days experimental trials in all the treatments groups except in Treatment 4 group (the antibiotic group) (Fig. 3D).

line 239- Add significance indication on figure 5

This comment is not clear to us

line 319- "Correspondingly, Bacteroidales was found in more abundance in all treatment groups; however, the levels decreased through days". FIGURE 1?

This is now rephrased as: Correspondingly, Bacteroidales are more abundant in all treatment groups; however, the number decreases through the 30 days of the experiments (Fig. 1B).

line 340- The phrase  "showing higher diversity than other treatment groups" was already said before

The statement is now deleted

line 341- "followed by Treatment 3 (combination of Lactobacillus rhamnosus and Enterococcus faecalis)" Is this significative? If is not you need to specify

It is not significant, and the statement is now added.

Round 3

Reviewer 3 Report

manuscript accepted